# Strong effect of demographic changes on Tuberculosis susceptibility in South Africa

**Oshiomah P. Oyageshio**[1☯]*, **Justin W. Myrick**[2☯], **Jamie Saayman**[3], **Lena van der Westhuizen**[3], **Dana R. Al-Hindi**[4], **Austin W. Reynolds**[5], **Noah Zaitlen**[6], **Eileen G. Hoal**[3], **Caitlin Uren**[3,7], **Marlo Möller**[3,7]*, **Brenna M. Henn**[1,2,4]*

1 Center for Population Biology, University of California, Davis, Davis, California, United States of America, 2 UC Davis Genome Center, University of California, Davis, Davis, California, United States of America, 3 DSI-NRF Centre of Excellence for Biomedical Tuberculosis Research, South African Medical Research Council Centre for Tuberculosis Research, Division of Molecular Biology and Human Genetics, Faculty of Medicine and Health Sciences, Stellenbosch University, Cape Town, South Africa, 4 Department of Anthropology, University of California, Davis, Davis, California, United States of America, 5 Department of Microbiology, Immunology, and Genetics, School of Biomedical Sciences, University of North Texas Health Science Center, Fort Worth, Texas, United States of America, 6 Department of Computational Medicine, David Geffen School of Medicine, University of California, Los Angeles, Los Angeles, California, United States of America, 7 Centre for Bioinformatics and Computational Biology, Stellenbosch University, Stellenbosch, South Africa

☯ These authors contributed equally to this work.
* oyageshio@ucdavis.edu (OPO); marlom@sun.ac.za (MM); bmhenn@ucdavis.edu (BMH)

**Data Availability Statement:** All R scripts for statistical data analysis and visualization are available at https://github.com/oshiomah1/NCTB-

## Abstract

South Africa is among the world's top eight tuberculosis (TB) burden countries, and despite a focus on HIV-TB co-infection, most of the population living with TB are not HIV co-infected. The disease is endemic across the country, with 80–90% exposure by adulthood. We investigated epidemiological risk factors for (TB) in the Northern Cape Province, South Africa: an understudied TB endemic region with extreme TB incidence (926/100,000). We leveraged the population's high TB incidence and community transmission to design a case-control study with similar mechanisms of exposure between the groups. We recruited 1,126 participants with suspected TB from 12 community health clinics and generated a cohort of 774 individuals (cases = 374, controls = 400) after implementing our enrollment criteria. All participants were GeneXpert Ultra tested for active TB by a local clinic. We assessed important risk factors for active TB using logistic regression and random forest modeling. We find that factors commonly identified in other global populations tend to replicate in our study, e.g. male gender and residence in a town had significant effects on TB risk (OR: 3.02 [95% CI: 2.30–4.71]; OR: 3.20 [95% CI: 2.26–4.55]). We also tested for demographic factors that may uniquely reflect historical changes in health conditions in South Africa. We find that socioeconomic status (SES) significantly interacts with an individual's age ($p = 0.0005$) indicating that protective effect of higher SES changed across age cohorts. We further find that being born in a rural area and moving to a town strongly increases TB risk, while town birthplace and current rural residence is protective. These interaction effects reflect rapid demographic changes, specifically SES over recent generations and mobility, in South Africa. Our models show that such risk factors combined explain 19–21% of the variance ($r^2$) in TB case/control status.

Epidemiology-Project. The relevant raw genetic data is deposited in the European Genome-phenome Archive (study accession number: EGAS00001007850). To maintain the privacy and anonymity of our study participants, and following our IRB-approved protocol, epidemiological data is available upon reasonable request. For access, please contact the Stellenbosch University Health Research Ethics Office at ethics@sun.ac.za and Dr. Marlo Moller at marlom@sun.ac.za.

**Funding:** This work was funded by NIH grant R35GM133531 to BMH. This work was also partially funded by the South African government through the South African Medical Research Council and the National Research Foundation (UID41744) to all members of DSI-NRF Centre of Excellence for Biomedical Tuberculosis Research: MM, CU, LW and JS. The content is solely the responsibility of the authors and does not necessarily represent the official views of the National Institutes of Health or the South African government. The funders had no role in study design, data collection and analysis, decision to publish, or preparation of the manuscript.

**Competing interests:** The authors have declared that no competing interests exist.

## Introduction

Tuberculosis (TB) is the world's leading cause of death due to infectious disease, currently greater than COVID-19 [1]. The causative agent, *Mycobacterium tuberculosis (M.tb)*, is an obligate intracellular pathogen mainly infecting the lungs, and sometimes other organs [2, 3]. Approximately 25% of the world's population is infected with *M.tb* and the annual death toll is similar to COVID-19 (~1.5 million deaths). South Africa is amongst the top 30 'high burden' countries coping with TB, TB/HIV co-infection, and multi-drug resistance or rifampicin-resistant TB (MDR/RR-TB). TB is South Africa's leading natural cause of death [4] with an extremely high prevalence (446/100,000, [5]) and accounts for 3.3% of all global TB cases [1]. The Northern Cape presently has the highest TB incidence in South Africa (ZF Mgcawu district 926/100,000 [5]), but the lowest HIV prevalence (7.1% vs 13.5% National average), including the lowest density of people living with HIV [6].

Determinants of active TB progression are multifaceted, including: genetics, nutrition, social and economic conditions, behavior, and sex-specific biology [1, 7, 8]. Initial *M.tb* infection is largely determined by exogenous factors, such as TB prevalence in the community, population density (e.g., prisons), and working conditions (e.g., mining, healthcare workers) [9–13]. The lifetime risk of progressing to active TB following infection is 10%. This risk is the highest within the first 5 years of initial infection and is typically considered to be mediated by the host's innate and cell-mediated immune system [9, 14]. Individual (i.e., host) factors, however, have also been shown to increase risk of progressing to active disease. These include HIV/AIDS, poor nutrition or low body mass index, indoor air pollution (e.g., cooking with wood and poor ventilation, smoking, alcohol abuse, diabetes mellitus, and intravenous drug use [1, 9, 14, 15]. Studies in India have shown undernutrition to be among the strongest determinants for TB risk [15]. In South Africa specifically, poor living conditions, unemployment, low SES, age and male gender, race, smoking, and marital status have all been identified as contributing to TB risk [16–21].

The extent of these determinants' effects can vary across and within populations, necessitating epidemiological studies in differing contexts and communities [8]. Compared to medium or high-TB-incidence countries, the effect sizes for alcohol abuse, homelessness, and intravenous drug use are stronger in low-incidence populations [22]. In South Africa, multilevel modeling approaches have shown that provincial [16] and community income inequality [18] have strong effects on TB incidence and progression, independent of individual-level risk factors.

HIV increases TB risk by 20-fold, the largest known risk factor for progression to active TB, and TB is the leading cause of AIDS-related deaths [23]. The effect of HIV on suppressing the host immune system can reactivate a latent *M.tb* infection and increase susceptibility to initial infection [14, 23, 24]. Despite HIV being the strongest TB determinant, other TB risk factors explain the majority of global TB cases [9]. In South Africa, 59% of people with TB on the National TB Programme (i.e. on TB medication) are co-infected with HIV [25]. However, South Africa's first national TB prevalence survey found that only 28% of people with TB were also people living with HIV (PLWHIV) [25], a finding underscoring the necessity to extend TB research to those living without HIV in high TB burden areas. At the provincial level in South Africa, HIV prevalence explains little of TB incidence ($r^2 = 0.036$) [26]. The Western and Northern Cape Provinces have among the highest TB incidence yet the lowest HIV prevalence [27].

Here, we present a TB case-control study characterizing the individual-level risk factors for TB progression among HIV-negative patients with suspected TB from the Northern Cape. The Northern Cape has the highest TB incidence but the lowest HIV prevalence and PLWHIV

density, and overall low population density, canonical risk factors do not appear to be driving the extraordinary incidence rates. To focus on factors other than immune suppression, we exclude PLWHIV from the analysis. Controls from our study sample are people with suspected TB from local health clinics who were microbiologically confirmed to be negative for active TB. Controls are assumed to either have been previously exposed to or infected with *M.tb* (i.e., LTBI). In South Africa, TB transmission is driven largely by community spread, rather than household contacts [28, 29]. Cases, in contrast, are people who have microbiologically confirmed active TB or self-report a past active TB episode. We test three separate models comprising common risk factors, as well as factors that may uniquely affect South Africa. We find that exogenous factors like SES, cohort age, and residence/birthplace have a strong effect on TB progression, often equal to or greater than endogenous factors like gender or smoking/alcohol. These results suggest further research into the causal mechanisms behind exogenous risk factors and opportunities for TB prevention are warranted.

## Methods

### Research ethics statement

This study has been approved by the Health Research Ethics Committee (HREC) of Stellenbosch University (N11/07/210A) and the Northern Cape Department of Health (NC2015/008). All participants were adults (18 years and older) and provided written informed formal consent. Authors Justin W. Myrick, Jamie Saayman, Lena van der Westhuizen and Marlo Möller had access to identifiable information about participants as they were directly involved in data collection or database management. Access to these records commenced on 26th January 2016, and is still ongoing as it is an integral part of the Northern Cape Tuberculosis Project (NCTB).

### Inclusivity in global research

Additional information regarding the ethical, cultural, and scientific considerations specific to inclusivity in global research is included in the Supporting Information (S1 Checklist).

### Study design and recruitment

Participants (18 years and older) provided written informed consent and were recruited from 12 community health clinics from the ZF Mgcawu district in the Northern Cape Province of South Africa from 26th January 2016 – 15th May 2017, and 11th December 2018 – 11th March 2020. Community health clinics are the front line for TB screening and treatment, visited by 87% of people who seek TB care [25]. TB nurses referred patients with suspected TB (with $\geq 2$ TB symptoms: cough for $\geq 2$ weeks, night sweats, weight loss, and fever $\geq 2$ weeks, or interaction with a TB index contact) and known TB patients to our on-site RAs. All study participants took a clinic-administered sputum GeneXpert Ultra test for active TB at the time of the study interview and provided saliva for genotyping. Clinic medical charts were accessed by a staff research nurse to record GeneXpert test results and verify HIV status and TB history.

### Case-control assignment

Cases and controls were assigned based on both the participant's medical charts and self-reported data (Fig 1). Cases include anyone with active pulmonary TB in their lifetime *and* that was HIV-negative. Thus, cases could be partitioned into 1) clinically confirmed active TB (n = 208) at the time of enrollment, and 2) self-reported past TB episode(s) (n = 166). GeneXpert results, diagnostic test date, TB strain (drug resistance), and TB medication regimen were

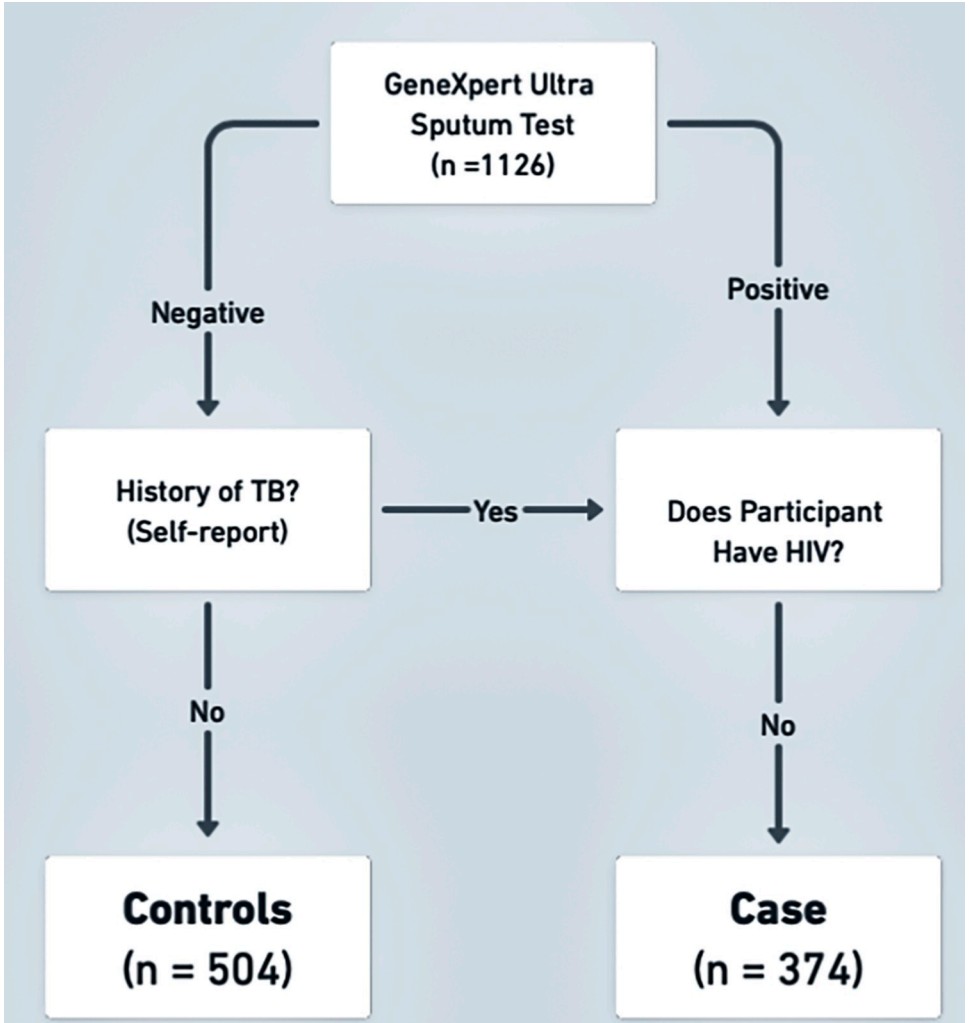

**Fig 1. Case-control decision tree.** Study participants were categorized as cases or controls based on medical record information and self-reported data. All participants were GeneXpert tested for active TB infection at the time of enrollment. Past TB episodes were self-reported and cross-referenced with medical records when available.

used to validate clinically confirmed progression to active TB. Past TB episodes are based on self-report, mainly due to older medical charts which were not reliably available, discarded, or difficult to locate by clinic staff.

We defined controls as HIV-negative clinic patients with suspected TB symptoms who had a negative GeneXpert Ultra result, and no history of active pulmonary TB at the time of study enrollment. Controls in our study are likely to be latently infected with *M.tb* (LTBI) or to have been exposed in their lifetime. A majority of the population in high TB burden South African suburbs are LTBI, 88% by ages 31–35 [30].

Our exclusion criteria removed participants with unknown TB or HIV status, as well as PLWHIV.

### Study covariates

We collected demographic information that included date of birth, place of birth, current residence, self-identified gender, self-reported ethnic identity, and parental ethnic identities.

Behavioral variables include smoking, drinking and diabetes (Supplementary Methods in S1 Text). In our analyses, we only used binary measures for smoking, drinking and diabetes ("Do you smoke?", "Do you have diabetes?"). Age at enrollment was used as a continuous variable for all analyses and binned for calculating empirical odds. Socioeconomic status "SES" was operationalized as number of years of education, i.e., the highest completed level of education. McKenzie et al. [31] have shown education level, in this dataset, positively predicts body mass index in TB controls, tracking access to resources and food security if only a crude measure.

Residence and birthplace locations are categorized as rural (≤2000 people) and town (>2,000 people). Population size was derived from the South African census and when census data was absent, e.g., a farm, we used Google Earth (earth.google.com) to estimate population size based on the number of dwellings. Places that did not have a census size available in Stats SA (statssa.gov.za) typically were very small communities, like a farm or a small settlement. By visualizing the settlement through Google Earth, we were able to estimate whether the community size was >2000 estimated people. We used the average household size from the census data from all locations in the district listed by Stats SA (statssa.gov.za), then counted the number of dwellings in blocks and multiplied by the average household size [4] to get an estimated population size. While an imprecise measure of population size, the lack of government census data for a community is itself an indicator of its rural locality.

## Statistical analyses

Statistical analyses were performed in R (version 4.2.3). We calculated Pearson correlations with the R package *ggcorrplot*. All categorical variables were numerically coded to "0" and "1". Classification models for our binary, qualitative dependent variable ("case"/ "control") included logistic regression and random forest—a machine learning classifier robust to non-linear associations and unknown variable interactions [32] (Supplementary Methods in S1 Text). To calculate empirical odds, we binned our participants into 7 age groups, dividing the number of controls by the number of cases in each age bin. All R scripts for analyses are available at https://github.com/oshiomah1/NCTB-Epidemiology-Project.

## Imputing missing data

To maximize our sample size, we imputed missing data for diabetes, smoking, and years of education (Table C in S1 Text). The proportion of missing data was overall low, i.e. below 5%, except for alcohol use at for which missingness was 35%; the alcohol use measure was implemented after a pilot study. We chose to exclude alcohol from imputation and model analyses due to high missingness. Multivariate imputation was performed using the R package MICE, implementing chained equations in which every variable with incomplete data is imputed conditional on all the data from other variables in the dataset [33]. To initiate the MICE procedure, we created a matrix of variables consisting of age, gender, height, HIV, mother's ethnicity, father's ethnicity, diabetes, smoking status, and years of education. Notably, ethnicity and height variables were not used in our regression analysis but bore potential relevance to our missing variables, so they were included to improve the statistical inference of our imputation. We set the parameters of our imputation using recommended settings [33, 34], generating two imputed datasets ($m = 2$) that were run for 10 iterations each (*maxit = 10*). We used a classification and regression tree method which is robust in epidemiological datasets similar to ours [35].

To cross-validate our imputation method, we randomly sampled ten percent of known values in each variable and converted them to missing values (Table C in S1 Text). Next, these missing values were imputed using the procedure described above, and the missing value was

compared to the original value. For continuous variables, the average percent difference between imputed and original value was used to calculate the cross-validation (CV) score while the average accuracy of the imputed variable was used to generate the CV score for binary variables. This procedure was carried out on one variable at a time for 100 iterations. Cross-validation results revealed that years of education and diabetes were sufficiently imputed (CV score > 10%).

## Obtaining and visualizing model coefficients

After MICE imputation, we used the 'psfmi' package [36] to implement our logistic regression models, obtaining pooled odds ratios using Rubin's Rules [37]. Each model was Bonferroni corrected using a baseline of $p<0.05$. For the Residence model, we set lifetime rural dwellers as the baseline and manually calculated contrasts for the other three comparisons. To illustrate the covariate effects from our models, we extracted the first imputed dataset from the MICE output, used the R 'glm' function to implement the logistic regression models, and then used the 'effects' package [38] to visualize the odds of Active TB.

## Genetic data processing & ancestry estimation

A subset of participants ($n = 159$) was genotyped for >2 million single nucleotide polymorphisms (SNPs) on the Illumina H3Africa array. Genetic data processing involved DNA extraction from saliva samples, common variant calling with GenomeStudio, rare variant calling with zCall, and further data cleaning using *plink2* (Supplementary Methods in S1 Text). Global [i.e. genome-wide] ancestry estimates were calculated using ADMIXTURE v1.13 [39]. The Luhya, Maasai, Himba, British, Palestinian, Chinese, Bangladeshi, Tamil, Ju|'hoansi San, Khomani San, and Nama populations were used as reference groups encompassing all major ancestry sources. ADMIXTURE was run in groups of maximally unrelated individuals to avoid biasing the ancestry estimates. We assumed k = 5 possible ancestries, inferred in unsupervised mode for each of the running groups. After matching clusters, we merged ancestry estimates across all running groups, averaging individuals that appeared in multiple running groups using pong [40]. We further tested whether population stratification affected the results of the logistic regression models by including 10 principal components (computed with *plink2*) and re-computing regressions for just the subset of $n = 159$ individuals (Supplementary Methods in S1 Text).

## Results

### TB case-control classification

1,126 participants were partitioned into preliminary cases, preliminary controls, and unverified TB status (571, 504, and 51 respectively). After excluding, participants with unverified TB status and either unknown or positive HIV status, 774 participants remained in the study (374 cases and 400 controls; Table A in S1 Text).

### Socio-behavioral and demographic characteristics of the cohort

Men and women were equally represented in the dataset (384 vs. 390, respectively, Table A in S1 Text). Men were more likely to drink alcohol (p < 0.001) and smoke (p < 0.05). A high fraction of our participants self-reported smoking (67%) and drinking alcohol (46%); smoking and drinking were moderately correlated with each other ($r = 0.36$, p < 0.05; Fig A in S1 Text). Women were more likely to have diabetes (p = 0.0004; Fig A in S1 Text) and, on average, had more education than men (female mean = 8.2 years, male mean = 7.8 years).

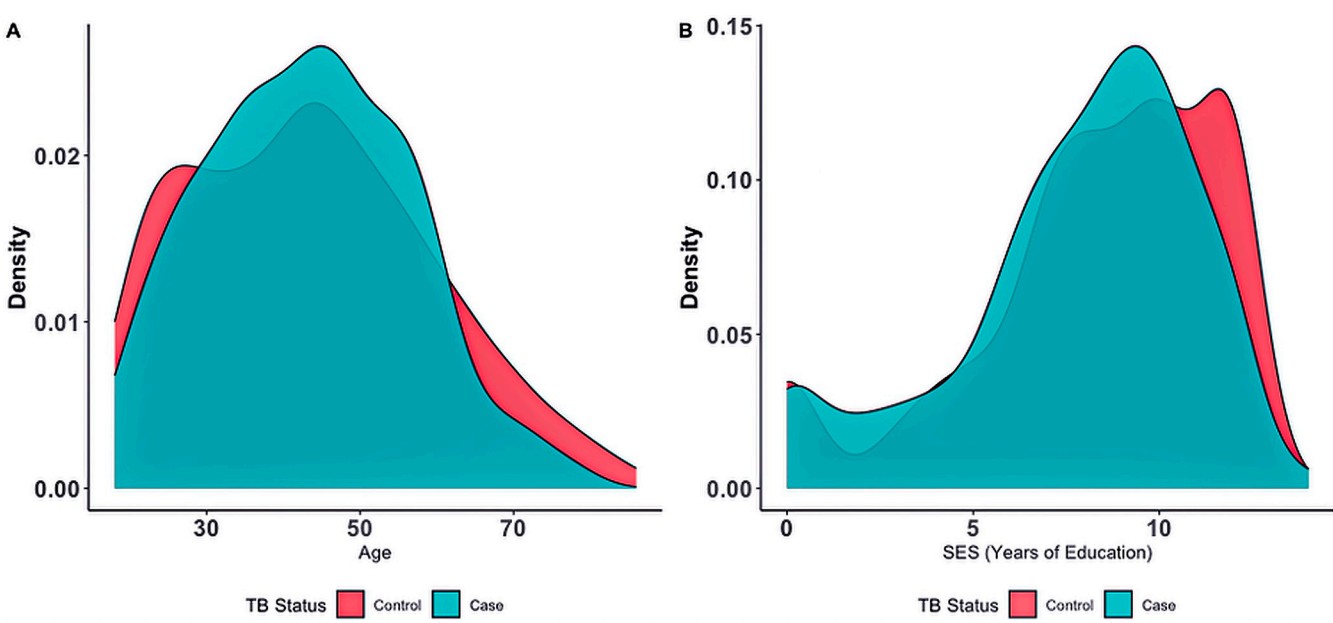

**Fig 2.** Density plots of continuous variables A) Age by case-control status B) SES by case-control status.

We use the number of years of education as a proxy for socio-economic status "SES" (*Methods*). The mean educational attainment in our cohort is 8 years, equivalent to completing primary school, and is similar between individuals recruited in rural areas and towns [ANOVA, p > 0.1). In the ZF Mgcawu District census [41], 13% of adults have not completed primary school as compared to 25.3% of our participants. Age was moderately correlated with SES (r = -0.5, p < 0.05; Fig A in S1 Text) such that older participants tended to have lower SES. Cases and controls had similar mean ages, 43.6 and 43.1 years respectively (Wilcoxon rank sum *p*-value = 0.959) (Fig 2A). We found a significant difference in SES between cases and controls, with mean of 7.7 years and 8.3 years, respectively (Wilcoxon rank sum *p*-value = 0.0019) (Fig 2B).

To investigate the possibility of selection bias we computed empirical odds of active TB by age group. Assuming that age is a cumulative measure of exposure (that is, capturing the amount of time someone is exposed to TB), the empirical odds of TB should increase monotonically with age. We observe a non-monotonic trend where the odds of active TB progressively increase from ages 18 up to 38, then reverses, progressively decreasing starting at age 39 up till the 79–88 age group having the lowest empirical odds (Fig B in S1 Text).

## Ethnicity and genetic ancestry

Individuals were asked to self-identify their ethnicity without categorical prompts. 88.4% of participants [both TB cases and controls) self-identify as Coloured, followed by 4.2% as a Khoe-San ethnicity (e.g., Nama, San), 4.6% as Tswana, 1.3% as Xhosa, and 1.9% as "other". Whilst we acknowledge that in some contexts the term "Coloured" has derogatory connotations, in South Africa it is a recognized ethnicity as well as a racial category. People who self-identify using this term tend to have genetic ancestries from multiple geographic origins, including the indigenous Khoe-San groups (e.g., Khoekhoe, San), Bantu-speaking, European, Indian, Malaysian (Southeast Asian) slaves, or people of mixed ancestry and their descendants [42]. The use of "Coloured" in this context reflects the self-identified cultural attributes of the

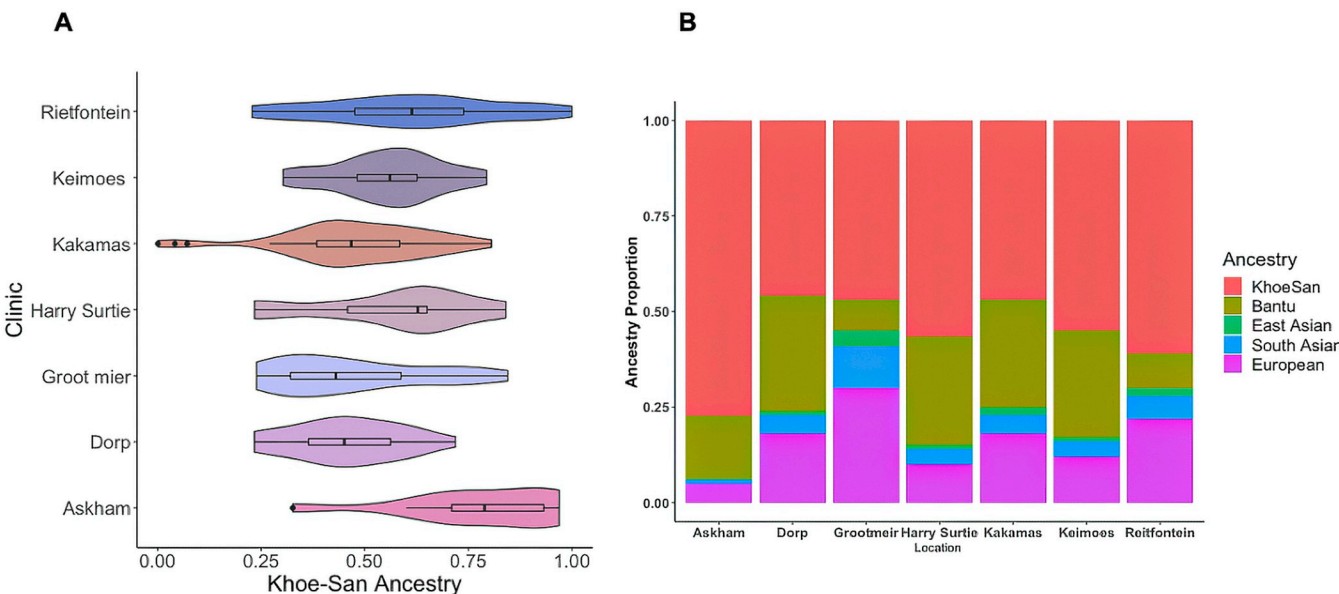

**Fig 3. Khoe-san ancestry is the primary genetic ancestry in clinics from the northern cape, South Africa.** A subset of participants (n = 159) was genotyped to obtain the average genome-wide ancestry proportions across all individuals for each clinic. A) Khoe-San ancestry is the largest proportion of ancestry in our sample, it varies significantly across study sites. The boxplots show the median, the 25th and 75th percentile and 1.5 times said percentile, and all outliers as dots. B) The study population is admixed with 5 distinct ancestries with the southern African indigenous Khoe-San ancestry being the highest proportion of ancestry at all study sites.

participants, as well as possible historical and genetic attributes. Ethnicity is reported out of respect for participants' choice of identity.

Genetic ancestry characterization was performed for 159 participants to assess if there was significant variation among clinics which could potentially confound analysis. Since genetic information was not available for all participants in the study, individual ancestry was not directly factored into the logistic regression models and random forest models. Khoe-San ancestry varied across clinic locations (Fig 3A) but remained the majority ancestry at each site (mean = 56%), followed by Bantu-speaking African ancestry (mean = 21%), European ancestry (mean = 16%), South Asian ancestry (mean = 5%), and East Asian ancestry (mean = 2%) (Fig 3B).

There was statistically significant variation in ancestry at the clinic level (ANOVA: Khoe-San, F = 6.9, p < 1e-06; Bantu-speaking, F = 7, p < 6e-05; European, F = 5, p < 6e-05; South Asian, F = 8, p < 4.8e-08; East Asian, F = 4.5, p < 0.001, Fig 3A). The statistically significant differences in the proportion of mean ancestry were generally between clinics in the Kalahari (Askham and Rietfontein, except Groot Mier) versus clinics along the Orange River (Harry Surtie, Dorp, Kakamas, and Keimoes) determined by Tukey HSD post hoc tests. The Kalahari clinics tended to have more Khoe-San ancestry (20–30% more) than the Orange River clinics and less Bantu-speaking African ancestry (~20% less). Notably, Groot Mier in the Kalahari had more European (11–19% more) and greater South and East Asian ancestry (4–10% and 2–3% more, respectively) than most other clinics (Fig 3B), likely due to Groot Mier's history as an early European colonial post [43].

## Hypotheses and models of progression to active TB

We designed three logistic regression models [44] to examine the risk factors that determine TB case/control status in our sample. Our first model, which we termed the "common risk

factor" model (n = 774), includes six covariates known to be common behavioral or demographic risk factors for TB. We hypothesized that risk factors identified from prior studies are also significantly associated with active TB progression in our HIV-negative population.

**Common risk factor model:** TB Status ~ gender + smoking + diabetes + residence + age + SES

Health disparities are one of the many consequences of apartheid in South Africa [45, 46]. The end of apartheid circa 1994 improved social mobility and educational access; however, health disparities in the Northern Cape (and other provinces) still remain problematic [47]. We formulated two alternative models which included variables potentially important to South African populations, involving the change in SES over time, and migration between rural and urban areas. We hypothesized that there are differential effects of SES (education) on TB status due to the sociopolitical effects of apartheid. To capture the effect of lived experience vis-à-vis Apartheid on TB outcomes we designed an "SES model" (n = 774). This model includes the common risk factors as above but allows for an interaction between age and SES to account changing economic conditions over the past eighty years. For example, completion of a high school equivalent education in 1960 did not afford the same economic benefits as completion of a high school education in 2010. We predict that for younger cohorts, higher SES is protective against TB; in contrast, for individuals born during apartheid, higher SES would have little effect on lifetime TB status. Age was kept as a continuous variable because apartheid was not a historically binary event.

**SES model:** TB Status ~ common risk factor model + age * SES

Residing in an urban or rural environment is an established risk factor for TB status [11, 48–50] and was included in our common risk factor model, as above. Previously, we have shown that migration from an individual's natal town has increased over the past two generations in the rural Northern Cape Province [31]. In addition, a recent longitudinal study leveraging data from South Africa's National Health Laboratory Service showed that incorporating cross-municipality migration improves the ability to predict TB incidence [51]. For our "residence model", we hypothesized that migrating from a rural to urban area in one's lifetime increases the odds of active TB status due to greater exposure to *M.tb*. Here, we include an interaction between current residence and birthplace in the common risk factor model. Setting this interaction allows us to examine four patterns, namely: lifetime rural residence, rural birthplace to urban residence, urban birthplace to rural residence, and lifetime urban residence.

**Residence Model:** TB Status ~ common risk factor model + residence * birthplace

The common risk factor model (pseudo $R^2$ = 19%, n = 774, Table 1) performed slightly worse than the SES model (pseudo $R^2$ = 21%, n = 774, Table 2). The residence model had a smaller sample size (pseudo $R^2$ = 19%, n = 720, Table 2) than the common risk factor model due to missing birthplace data for some individuals. For an equal comparison, we re-ran the common risk factor and SES models with same individuals as in the Residence model. For the reduced dataset the Residence model and SES model had comparable pseudo $R^2$ while the common risk factor model had a slightly worse value (pseudo $R^2$ = 18% vs. 19%) (Table B in S1 Text). Therefore, we present results from all three models and contrast the variable effects (Tables 1 and 2). All significance levels were Bonferroni corrected, assuming an $\alpha$ = 0.05.

## Common risk factors

Across the three models, males consistently have three times the odds of active TB than females (OR = 3.01 [2.20,4.12], $p < 0.001$; Tables 1 and 2, and Fig C in S1 Text). All logistic regression models showed insufficient statistical evidence for smoking (common risk factor

**Table 1. Odds ratios and *p*-values for the demographic and socio-behavioral variables used in the common risk factor model.**

|  | Common Risk Factor (n = 774) | |
|---|---|---|
|  | **Odds Ratio (95% CI)** | **p value** |
| **(Intercept)** | 0.23 [0.08, 0.66] | 0.007 |
| **Residence** | 3.20 [2.26, 4.55] | p < 0.001** |
| **Gender** | 3.01 [2.20, 4.12] | p < 0.001** |
| **Age** | 0.99 [0.97, 1.00] | 0.046 |
| **Years of education (SES)** | 0.91 [0.86, 0.97] | 0.006* |
| **Smoker** | 1.32 [0.94, 1.85] | 0.111 |
| **Diabetes** | 1.27 [0.67, 2.43] | 0.462 |
| **n** | 774 | |
| **R²** | 0.19 | |

\* alpha = 0.05

\*\* alpha = 0.01

model: OR = 1.32 [0.94, 1.85], *p* = 0.11; Tables 1 and 2), and diabetes (common risk factor model: OR = 1.27 [0.67, 2.43], *p* = 0.46; Tables 1 and 2) on TB risk. Despite the lack of significance, we note that smoking had an effect size in the expected direction (Fig C in S1 Text). The variable with the strongest effect size was current residence in towns–areas with a population size greater than 2,000 peoples (OR = 3.20 [2.26, 4.55], *p* <0.0001; Tables 1 and 2 and Fig C in S1 Text).

## Age interacts with SES

In the common risk factor model, age does not significantly affect TB risk, but higher SES has a protective effect (OR = 0.91 [0.86, 0.97], *p* = 0.006; Table 1). In the SES model, SES significantly affects TB status depending on age group (OR = 1.01 [1.00, 1.01], *p*< 0.001, Table 2).

**Table 2. Odds ratios and *p*-values for the demographic and socio-behavioral variables used in the residence model and SES model.**

|  | Residence Model | | SES Model | |
|---|---|---|---|---|
|  | **Odds Ratio (95% CI)** | **p value** | **Odds Ratio (95% CI)** | **p value** |
| **(Intercept)** | 0.36 [0.12, 1.12] | 0.077 | 3.69 [0.50, 27.36] | 0.197 |
| **Birthplace^** | 2.84 [1.28, 6.21] | 0.005* |  |  |
| **Residence^** | 6.10 [2.97, 12.5] | p <0.001** | 3.29 [2.30, 4.71] | p <0.001** |
| **Gender^** | 3.04 [2.19, 4.21] | p <0.001** | 3.02 [2.20, 4.15] | p <0.001** |
| **Age** | 0.98 [0.97, 1.00] | 0.022 | 0.94 [0.91, 0.97] | p <0.001** |
| **Years of education (SES)** | 0.92 [0.86, 0.98] | 0.013 | 0.66 [0.54, 0.81] | p <0.001** |
| **Smoker** | 1.32 [0.93, 1.89] | 0.124 | 1.22 [0.85, 1.73] | 0.274 |
| **Diabetes** | 1.49 [0.74, 3.00] | 0.267 | 1.28 [0.67, 2.45] | 0.453 |
| **Years of education: Age** |  |  | 1.01 [1.00, 1.01] | p <0.001** |
| **Birthplace: Residence** | 0.33 [0.13–0.86] | 0.024 |  |  |
| **n** | 720 | | 774 | |
| **R²** | 0.19 | | 0.21 | |

\* alpha = 0.05

\*\* alpha = 0.01

^ We report the OR with a baseline of rural = 1, for Birthplace; a baseline of town = 1, for Residence; for Gender the OR is listed for males, and for Smoking and Diabetes the OR is listed for a positive response.

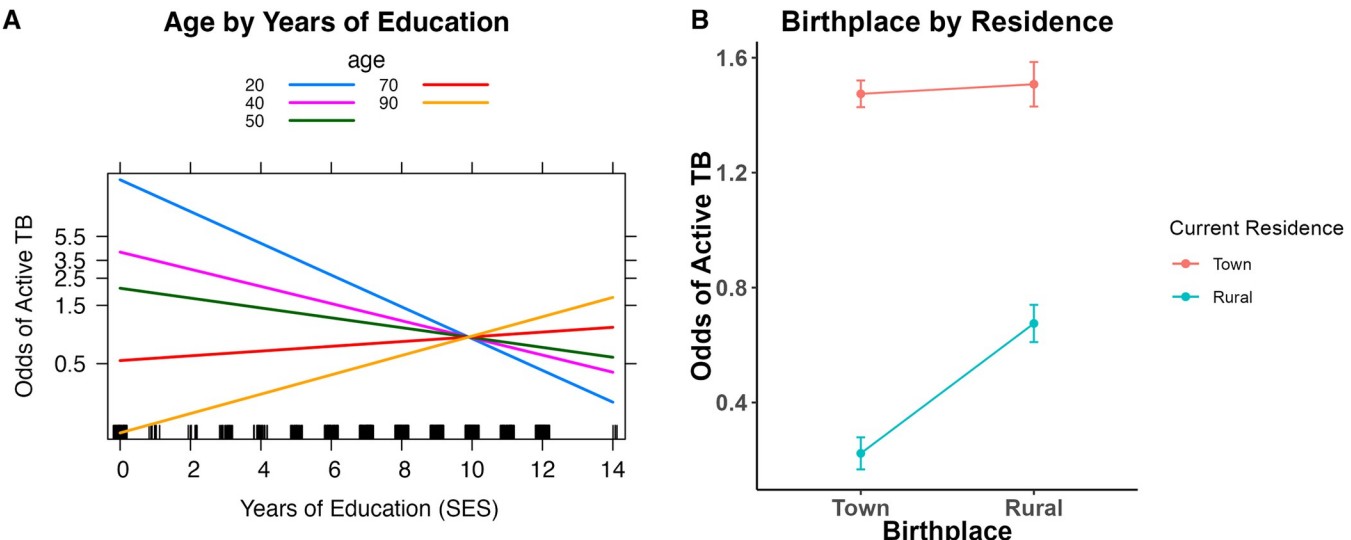

**Fig 4. Logistic regression interaction plots.** A) The odds of active TB by education level vary across age groups (shown above by the different color lines). More years of education decreases the odds of active TB in younger age groups, but this pattern reverses in the oldest age groups. B) Effect plot from the residence model visualizing an interaction term between birthplace residence and current residence. Regardless of birthplace, the odds of active TB is highest in individuals who currently reside in towns. Individuals born in towns and currently residing in rural areas have the lowest odds of active TB.

The effect is such that higher SES at younger ages (18–59 years old) is protective against TB, and higher SES at older ages (>59 years) increases risk (Fig 4)

## TB risk by residence and birthplace

We analyzed the relationship between TB status and a change of locality between birthplace (rural area or town) and current residence (rural area or town) during an individual's lifetime. We expected to see a difference in the odds of active TB between lifelong residents and those who have moved between locales. Under such a model, lifelong rural dwellers would have the lowest odds and lifelong town dwellers would have the highest odds. We set an interaction term between current residence and birthplace classified into town/rural, (OR = 0.33 [0.13–0.86], p = 0.024, Table 2). To break down the interaction effect, we set lifelong residents of rural areas as the baseline, comparing the three other residence patterns with this baseline group. We found that lifelong town dwellers had about twice the odds of active TB (OR = 2.16 [1.43–3.28], $p < 0.001$) relative to the baseline. Individuals born in a rural area and currently residing in a town had similar outcomes as lifelong town dwellers (OR = 2.19 [1.14–4.20], $p = 0.018$). Taken together, these show that town residence increases risk regardless of birthplace. Interestingly, individuals who were born in a town and later moved to rural areas are even more protected than individuals born and currently residing in rural areas (OR = 0.33 [0.16–0.71], $p = 0.004$) (Fig 4).

## Random forest modeling

As an alternative to logistic regression, we trained a random forest model to classify TB status utilizing all the variables from the common risk factor model. We configured the model to grow 5000 classification trees (Supplementary Methods in S1 Text). The model assigned gender, current residence, SES, and age as the top important independent variables (Fig 5). Diabetes and smoking were classified as uninformative predictors for TB status. The model had an overall "out-of-bag" misclassification rate of 23%.

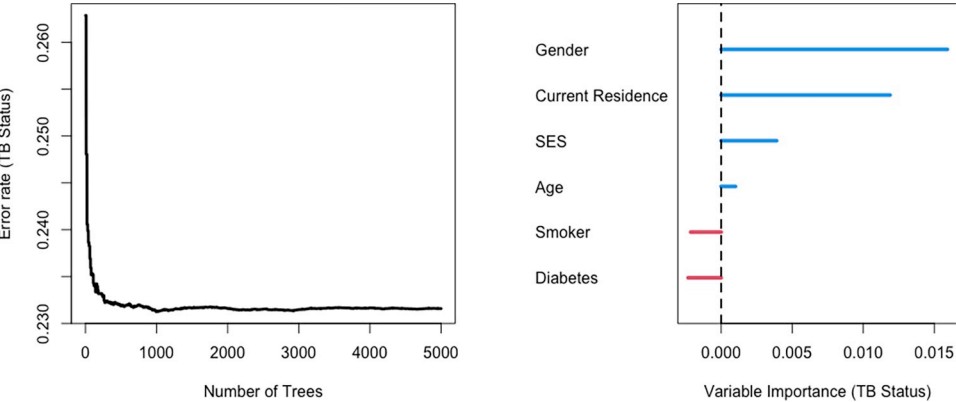

**Fig 5. Random forest model with common risk factor variables (n = 774).** Random subsets of all 6 variables on the y-axis were used to grow 5000 trees to classify participants into cases and controls. The model had an overall "out-of-bag" misclassification rate of 23%. Variables with higher variable importance are most crucial for case-control classification. Predictor variables with negative variable importance values worsen the ability of the model to classify TB status.

## Discussion

We examined common demographic risk factors for TB, constructing the largest TB epidemiological study in a Northern Cape clinical population (n = 774), to our knowledge. We show that gender, SES, and current residence locality are significant variables or important TB risk predictors, using logistic regression and random forest models. Neither smoking nor diabetes is associated with increased TB risk in any model. Among the three logistic regression models, interacting SES by age ("SES model"), and birthplace by residence ("residence model"), had similar explanatory power, improving on the common risk factor model.

In South African townships, *M.tb* is community spread [28, 29] and about 88% of adults are infected by ages 31–35 [30]. Here, we demonstrate the utility of sampling in high disease incidence populations to rapidly build datasets with large sample sizes of TB cases and controls with similar pathogen exposure. Validating *M.tb* exposure in controls is traditionally done by a tuberculin skin test (TST) and/or interferon-gamma release assay (IGRA; e.g., Quanti-FERON), thereby assigning LTBI status. We did not perform IGRA and TST tests for all controls in our cohort because IGRA testing requires blood draws and is often prohibitively expensive for large cohort studies, and TST is not readily available in South Africa. To validate the LTBI rate in our controls, we IGRA-tested (within 3 days of the participant's GeneXpert result), a random selection of our sample controls (n = 70) and found that they have an 87% LTBI rate (IGRA+). The highly significant model results, in combination with the IGRA+ subset of controls, suggest that our sampling strategy reliably categorizes TB disease risk.

We considered whether our sampling strategy displayed any indication of selection bias. We hypothesized that the risk of active TB should monotonically increase with age, reflecting cumulative lifetime exposure to the *M.tb* pathogen. Contrary to this, our findings revealed a non-monotonic relationship. Starting with the youngest age group (18–28 years), the empirical odds of active TB increased with age, peaking in the 29-38-year-old age group, followed by a progressive decline in empirical odds till the oldest age group (79–88 years) (Fig B in S1 Text). This unexpected pattern can be indicative of selection bias, possibly driven by survivor bias—where individuals dying from TB are absent from the study population at older age bins and/or related to whether our controls are representative of the general population (sample-selection bias). Generally, selection bias is difficult to measure and mitigate [52–54], especially in

case-control studies where controls are recruited from a clinical setting. Recall bias in older adults could also lead to the lower observed empirical OR in older age bins; however, because TB treatment is a six-month course, and incomplete treatment regimens lead to relapse or life-threatening drug resistant TB, it is generally a life event that people remember with the exception of pediatric TB.

## SES model effects

Age-specific TB risk varies across the lifespan. The greatest risk of TB is during infancy, decreasing through adolescence, then increasing and peaking between 25–35 years old followed by a decrease, and another peak after 65 years [48, 55]. Age was not a significant variable in the logistic regression except when interacting with SES. SES's protective effect on TB risk is most evident among 18–39-year-olds but the trend reverses and increases risk among the eldest individuals (>69 years; Fig 4A)—those who grew up and reached adulthood during Apartheid (Fig 4A). Higher SES increasing TB risk at older ages is contrary to findings in populations in the United States and Mexico [55]. This unique pattern may reflect South Africa's recent history of Apartheid and post-Apartheid societal and economic shifts. During Apartheid, individuals from historically marginalized backgrounds had limited career options, but some were able to become teachers, police officers, or nurses. Such occupations are associated with higher education requirements and would have facilitated access to larger salaries, transportation, and mobility; potentially leading to better diagnosis and treatment. Alternatively, the observed pattern of our interaction effect at older ages could be explained by selection bias or our operationalization of SES in this study. Highest completed level of education (e.g., grade, diploma, degree, etc.) is a blunt measure of SES, and does not fully capture all SES facets, including social, economic, and cultural capital [56, 57], and universal access to education increased post-Apartheid [58]. Additionally, we only sample from community health clinics, not private clinics, thereby missing a fraction of the SES spectrum.

## Residence model effects

Consistent with previous research [21, 59–61], we find TB risk is associated with living in larger towns. In our prior work, mobility in the Northern and Western Cape populations changed over the past 3 generations, with the highest levels of mobility in the grandparental generation [62]. Therefore, we tested whether mobility (operationalized as a different birthplace and residence) affected TB risk. Individuals currently residing in towns (regardless of birthplace) had higher odds of active TB, compared to those born in towns that migrated to rural areas, and lifetime rural dwellers. Unexpectedly, the individuals with the lowest odds of active TB are those born in a town who move to a rural area (Fig 4B). When we returned results to the community, the clinic staff hypothesized that despite nationally standardized BCG vaccinations, rural areas may have lower vaccination rates (observation communicated by clinical staff in the study catchment), therefore those born in town benefit from a greater likelihood of greater BCG vaccine during childhood and low adult *M.tb* exposure living in rural areas. The benefits of the BCG vaccine, however, attenuate in adolescents and memory of childhood TB episodes suffer from recall bias. Another possibility is that the town-born and rural-residence group accrued more wealth in towns before moving to a rural area, affording a different lifestyle than their rural neighbors (e.g., afford larger homes, less crowding, cleaner cooking fuel, etc.). This unique combination of factors may explain why the town-born rural-residence group has even lower odds of active TB compared to the lifetime rural dwellers. Future work should consider collecting birthplace in addition to current residence to better identify TB risk as *M.tb* exposure varies across the lifespan.

## Common risk factors

Across global studies, men are on average 1.7 times more likely to have TB [63–65]. Sex biases like this are common in other infectious diseases [66, 67] and are attributable to an intersection of sex (biological factors, e.g., immune function) and gender (social and behavioral factors, e.g., risk-taking behavior) [68]. Despite smoking not being a significant TB risk, we found 75.5% of men smoke compared to 55.8% of women, indicating at least some gender differences in risky behaviors in the Northern Cape population.

Smoking and alcohol consumption has been shown to increase TB risk and mortality in the Northern Cape and at the national level [18, 69–71]. In our models, smoking had the expected effect on TB risk and alcohol was excluded from our models due to high missingness. Self-reporting biases in observational studies like this one are a concern for variables like smoking, alcohol consumption, and SES measures [72]. Our sample, however, reports much higher levels of smoking compared to large-scale national surveys (e.g., [73], men: 75.5.% vs. 41%; women: 55.8% vs. 21%, respectively suggesting minimal self-report bias in our study. The weak effect of smoking observed from our models may be due to our method of binary classification. We collected fine-scale smoking phenotypes (Supplementary Methods in S1 text) but because of the high missingness of these phenotypes, we ultimately classified participants as Smokers/Non-smokers. This stratification may mask the heterogeneity of smoking behaviors such as casual and binge substance use or differences in the types of smoking materials consumed.

## Ancestry & ethnicity

Finally, we highlight that our study included enrollment from 12 different clinics, some of which are more than 250 kilometers apart. We surveyed ethnicity and genetic ancestry to test for population structure in the sample. Such structure can confound analysis if genetic ancestry tracks differential host risk for progression to TB or if different ethnicities have different cultural norms. Previous studies have described the high proportion of Khoe-San ancestry in Northern Cape communities but these largely focused on descendant groups who identify as Khoe-San (e.g. the ≠Khomani San, the Nama, Karretjie) rather than the general population [74]. Here, we show the clinical study population to be admixed with 4 other distinct ancestries (Fig 3), demonstrative of recent historical events. These include the Bantu expansion into Southern Africa, European colonialism, the Dutch East India Company (aka VOC] slave trade, and the displacement and forced settlement of indigenous South African Khoe-San groups, especially in the last few generations in the Northern Cape [43, 75]. Although we do observe heterogeneity in ancestry across clinics, correcting for the top 10 genetic principal components did not change the logistic regression results (Fig D in S1 Text and S1 Data). To our knowledge, this is the first study to report ancestry proportions of clinical populations in the Northern Cape Province, South Africa. This work provides a baseline to design future studies, such as exploring host genetic correlates of active TB progression in this population (Supplementary Discussion in S1 text).

## Conclusion

Active TB progression is a multifactorial process involving the environment, genetics, and their interaction [1, 7]. Our results from the NCTB cohort indicate that sociodemographic variables strongly impact active TB risk. Effects that are unique to the Northern Cape Province may reflect how changes in the pre- to post-apartheid environment modified social factors, such as SES and mobility, which in turn impacted lifetime TB risk.

## Supporting information

**S1 Checklist.**
(DOCX)

**S1 Data.**
(XLSX)

**S1 Text.**
(DOCX)

## Acknowledgments

We would like to thank all the participant communities in the Northern Cape for their continued trust and support in helping us undertake this project. We would also like to thank our community research assistants and translators who assisted in data collection for the project. We are grateful to Prof. Faadiel Essop, Dr. Desiree Petersen, Prof. Eileen Hoal, and Prof. Leslie Swartz for closely reading this manuscript. We would also like to thank Dr. Chris Gignoux and Dr. Mark Grote for statistical advice. Finally, we want to thank the Department of Health in the Northern Cape Province, South Africa for their continued support of the project.

## Author Contributions

**Conceptualization:** Oshiomah P. Oyageshio, Justin W. Myrick.

**Data curation:** Oshiomah P. Oyageshio, Justin W. Myrick, Jamie Saayman, Lena van der Westhuizen, Dana R. Al-Hindi, Brenna M. Henn.

**Formal analysis:** Oshiomah P. Oyageshio, Noah Zaitlen, Brenna M. Henn.

**Funding acquisition:** Eileen G. Hoal, Marlo Möller, Brenna M. Henn.

**Investigation:** Oshiomah P. Oyageshio, Justin W. Myrick, Austin W. Reynolds, Noah Zaitlen, Caitlin Uren, Marlo Möller.

**Methodology:** Oshiomah P. Oyageshio, Justin W. Myrick, Brenna M. Henn.

**Project administration:** Justin W. Myrick, Jamie Saayman, Lena van der Westhuizen, Dana R. Al-Hindi, Austin W. Reynolds, Eileen G. Hoal, Caitlin Uren, Marlo Möller, Brenna M. Henn.

**Resources:** Lena van der Westhuizen, Austin W. Reynolds, Eileen G. Hoal, Marlo Möller, Brenna M. Henn.

**Supervision:** Justin W. Myrick, Jamie Saayman, Caitlin Uren, Marlo Möller, Brenna M. Henn.

**Validation:** Oshiomah P. Oyageshio, Justin W. Myrick.

**Visualization:** Oshiomah P. Oyageshio.

**Writing – original draft:** Oshiomah P. Oyageshio, Justin W. Myrick, Marlo Möller, Brenna M. Henn.

**Writing – review & editing:** Eileen G. Hoal, Caitlin Uren.

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
