## [Decision Letter · Decision Letter 0]

8 Jan 2024

PGPH-D-23-02135

Strong Effect of Demographic Changes on Tuberculosis­­ Susceptibility in South Africa

Dear Dr. Oyageshio,

Thank you for submitting your manuscript to PLOS Global Public Health. After careful consideration, we feel that it has merit but does not fully meet PLOS Global Public Health’s publication criteria as it currently stands. Therefore, we invite you to submit a revised version of the manuscript that addresses the points raised during the review process.

Both reviewers have major issues related to how ancestry is used to draw conclusions in this analysis; the selection of cases and controls (specifically pertaining to HIV status) and subsequent biases; the validity of methods used to determine TB disease history; and the statistical analyses. Please address all major and minor comments in your revised submission.

We look forward to receiving your revised manuscript.

Kind regards,

Indira Govender, MBChB, MMed, FCPHM

Academic Editor

Journal Requirements:

Additional Editor Comments (if provided):

Reviewers' comments:

Reviewer's Responses to Questions

**Comments to the Author**

1. Does this manuscript meet PLOS Global Public Health’s publication criteria? Is the manuscript technically sound, and do the data support the conclusions? The manuscript must describe methodologically and ethically rigorous research with conclusions that are appropriately drawn based on the data presented.

Reviewer #1: Partly

Reviewer #2: Yes

2. Has the statistical analysis been performed appropriately and rigorously?

Reviewer #1: No

Reviewer #2: Yes

3. Have the authors made all data underlying the findings in their manuscript fully available (please refer to the Data Availability Statement at the start of the manuscript PDF file)?

Reviewer #1: No

Reviewer #2: Yes

4. Is the manuscript presented in an intelligible fashion and written in standard English?

Reviewer #1: Yes

Reviewer #2: Yes

5. Review Comments to the Author

Reviewer #1: This study focuses on the factors that affect TB risk in an understudied TB endemic region. The authors use a case-control study and find that the following factors increase TB risk: male gender, age x socioeconomic status, and birthplace x residence locality. The variables included in the interactions did not have main effects on TB risk, nor did smoking and alcohol consumption. Finally, the authors performed genotyping and report that participants had a majority Khoe-San ancestry, but genetic ancestry is never linked to TB risk. This paper represents a large amount of collaborative and interdisciplinary work and an important description of an understudied population. My main comment is that several parts of the analyses and decisions behind them are unclear, including whether and how genetic ancestry is analyzed in the context of TB.

Major comments:

The introduction discusses how genetic variation and genetic ancestry can impact TB risk, and line 91 says that ancestry proportions are investigated. However, I didn’t see these analyses anywhere in the paper.

Line 259 – Do you examine ancestry effects on host susceptibility? This concept is brought up again but no analyses are described in the methods.

Are there any relatives in your study? Or population structure? If so, please consider linear mixed effects models that control for pairwise genetic relatedness and/or principal components of genetic variation. Since you have genetic data, I’m not sure why it isn’t used for data analysis either to allow for modeling of a key covariate and/or to examine ancestry effects on susceptibility.

Why are HIV positive individuals not considered cases? They still have TB. From the flow diagram in Figure 1 it also looks like HIV positive status does not exclude someone from being a control, which seems like it would create bias if they are excluded from one group and not the other.

Minor comments:

Line 52 – “The extent of these determinants’ effects varies across and within populations, necessitating epidemiological studies in differing contexts and communities”. I’m sure this is true, but a few examples would really help. In general, the motivation for the particular study (beyond studying an understudied population) could use some fleshing out.

Can you provide some more background about why TB/HIV co-infection is so common? Is the thought that individuals succumb to TB after being infected with HIV because of compromised immune systems? It is also confusing that the intro focuses on TB/HIV co-infection, and then these individuals are excluded from the actual study.

It is very inconvenient to have the figure legends embedded in the text, but the figures themselves at the end of the manuscript.

It’s not clear until the methods that the study is really comparing suspected/latent TB individuals vs individuals with currently active or past active TB. Could you make that clear in the intro and reframe some of the background about risk factors for progression to active TB (rather than risk factors for TB alone)?

Line 159-161 – Please expand on this procedure. How were Google Earth images translated into population size? It would also be nice to compare censuses data to your Google Earth method for a few regions that have census data, to see how correlated the methods are and if there is any bias. Also, this part of the methods makes it sound like you gather population density info, but then there’s not explanation of how this is binarized into the variable you use for modeling (town vs rural).

Line 219 - Please justify the use of the particular populations included as reference groups in the ADMIXTURE analyses. Also why was k=5 chosen? Were other values of k tested?

Line 290 – Why not run the common risk factor model for the individuals included in the residence model? That way you could ask whether the inclusion of residence variables improves the model. As it stands, it’s very unclear because there is no base model to compare to.

Table 1 – You don’t need a blank row for “Current Residence - Rural (reference)”

Apologies if I missed it, but could you provide some ideas in the discussion about the interaction between birthplace and current residence, since the effects go in the opposite direction of what was expected?

Reviewer #2: This study aims to assess risk factors for TB in a sample of patients attending health facilities in the Northern Cape province of South Africa. Consistent with previous studies, male sex and lower educational attainment were found to be risk factors, but in further analysis the effect of educational attainment was found to differ substantially with respect to age. Also somewhat surprisingly, although there was a substantially higher TB risk in town dwellers, people who had been born in towns and resided in rural areas had the lowest TB risk.

Major comments

The study design is unusual because the “cases” are defined as people who either have current TB (microbiologically confirmed by GeneXpert testing) or who self-report having had TB previously. The latter is less ideal because self-report is generally not considered very reliable (there may be recall bias, particularly if the TB episode occurred a very long time ago, or there may be social desirability bias given the stigma associated with TB). The authors don’t report what fraction of the cases were self-reported versus microbiologically diagnosed, and so it’s difficult to know how serious this limitation is. This information is also important in assessing how much of the TB is past versus current TB. Did the authors assess whether there were differences in the TB risk factors when considering separately the current TB cases and the self-reported past TB cases?

There are a number of places in the paper where the authors allude to historical changes (or in some cases “demographic changes”) as possible explanations for interactions between risk factors, but without providing coherent explanations or clear hypotheses. For example, in lines 377-9: “This study demonstrates a possible unique historical context to South Africa, (post-)Apartheid differential effects between sociodemographic and health outcomes.” It is difficult to make a convincing argument when past TB and current TB are analysed as a single outcome.

Related to the previous point, the authors could do more to compare their study with other South African studies on the role of socio-economic status. It’s difficult to know whether the risk factors they describe as “different from other global populations” (in the abstract) are really a peculiarity of this study design/setting or consistent with other South African data. As examples of South African studies that have assessed the effect of socio-economic status on TB risk, see Ncayiyana et al (BMC Infectious Diseases, 2016, 16:661), Mahomed et al (Int J Tuberc Lung Dis, 2011, 15(3):331-6) and Harling et al (Social Science & Medicine, 2008, 492-505). It should also be acknowledged that educational attainment may be a crude measure of socio-economic status (especially in the post-Apartheid context of universal access to education). In addition, participants were recruited from public health facilities, and so people using private healthcare are not represented, which obviously limits the ability to draw conclusions about TB risk at higher SES levels.

The finding that TB risk is lowest in people who were born in towns but now reside in rural areas is a bit unexpected. The authors attempt to provide an explanation for this is the Discussion, arguing that BCG vaccine coverage is probably better in urban areas than in rural areas, meaning the town-born individuals are more protected early in life. But this argument seems a bit tenuous, given that the authors don’t cite any evidence of low BCG vaccine coverage in rural areas, and given that the BCG vaccine is generally thought to provide only limited protection in early childhood (a life stage that most people are unlikely to remember when asked if they’ve ever had TB). Another explanation might be that the people who were born in towns but now reside in rural areas are generally of higher socio-economic status.

The decision to exclude people living with HIV (PLWH) from the cases but not the controls seems a bit odd, and the rationale isn’t clear in the main paper. In the supplementary materials it becomes clear that the authors are trying to focus on non-HIV risk factors (it would perhaps have helped to explain this in the main text). But if one is going to exclude PLWH, does it not make sense to exclude them from both the cases and the controls? Describing HIV-positive non-TB patients as “resisters” (in the supplementary materials) is a bit weak, since the authors haven’t done any TST/IGRA testing, but even if it is true, how does this justify the decision to include them as controls?

The interpretation of Figure 2 could be questioned. The authors interpret the (relatively) low TB risk at older ages as indicating a “survivor bias”. Although that is possible, one should also consider that (a) these are self-reported data that could be affected by recall bias (more important in older adults), and (b) these are data from patients attending health facilities, and not a population-based sample (one might expect that older adults would be more likely to be attending health facilities because of chronic conditions). I did also wonder whether it was really necessary to include Figure 2 in the paper (and if so, whether it should not go in the Results section, rather than the Methods section).

A limitation is that 35% of the alcohol data in this study were imputed. Although this is mentioned in the supplementary materials, it’s important enough that it should be mentioned in the main text. The high level of missing alcohol data cou

---

## [Decision Letter · Decision Letter 1]

3 Apr 2024

PGPH-D-23-02135R1

Strong Effect of Demographic Changes on Tuberculosis­­ Susceptibility in South Africa

Dear Dr. Oshiomah Oyageshio,

Thank you for submitting your manuscript to PLOS Global Public Health. After careful consideration, we feel that it has merit but does not fully meet PLOS Global Public Health’s publication criteria as it currently stands. Therefore, we invite you to submit a revised version of the manuscript that addresses the points raised during the review process.

We appreciate the authors having made significant progress in addressing the first round of comments from reviewers. However there remain major concerns around the selection and description of controls, and the availability of data used in this analysis. Please note that due to a technical issue, reviewer 1 was only able to mark the revisions as acceptable but has provided feedback requesting minor revisions which are inserted below:

The authors have done a thorough job of addressing my comments. I appreciate the attention to detail and found the responses to be generally robust.

I have two minor outstanding comments:

Line 219 - Please justify the use of the particular populations included as reference groups in the ADMIXTURE analyses. Also why was k=5 chosen? Were other values of k tested?

● We have previously analyzed genetic ancestry data for other populations from the Western and Northern Cape (e.g. Uren et al 2016, Genetics; Petersen et al. 2013, PLoS Genetics; De Wit et al 2010, Human Genetics). Based on the observation of ancestry proportions we know that there are two ancestries (eastern Asian: Indonesian, Malay, Chinese; southern Asian: Sri Lanka, India, Bangladesh) which are minority ancestries typically less than 5% derived from the VOC slave trade at lower fractions compared to Cape Town. European migration is well documented historically and in interviews. Interviews also suggest that recent migration from Bantu-speakers impacts the communities, including Xhosa, Tswana and others. Finally, the Khoe-San ancestry could have been broken down into 2 clusters based on some dierentiation between Khoekhoe and San-derived demographic histories. But this ancestry dierentiation estimate tends to be unstable in ADMIXTURE, likely due to small samples sizes for the Khoekhoe. Hence, k=5.

Additional reviewer response: Got it. Please incorporate this information into the methods so it clear to all readers.

Are there any relatives in your study? Or population structure? If so, please consider linear mixed effects models that control for pairwise genetic relatedness and/or principal components of genetic variation. Since you have genetic data, I’m not sure why it isn’t used for data analysis either to allow for modeling of a key covariate and/or to examine ancestry effects on susceptibility.

●  We do see moderate population structure across clinics. Please see Supplementary section “Genetic ancestry and ethnicity” for detailed discussion.  

●  We agree it would be appropriate to include PC covariates or GRM for this analysis. It is the goal of this project is to perform these association analyses in the future with more data, however, we have only genotyped ~159 samples, which will be underpowered to identify genetic associations. Grant applications to genotype the full dataset have not yet been funded and would total ~$60,000 prohibiting us from including them in the current paper.  

Additional review response: Thanks, that makes sense and I understand the financial limitations for genotyping the whole cohort. However, 159 is a reasonable sample size for checking whether your scientific conclusions are likely to change when you account for population structure, which you demonstrate is pretty substantial. For example, I think it would be worthwhile to check if effect sizes dramatically change in magnitude or direction when you include PCs in the model using the reduced dataset. Or potentially you could ask about average TB patterns as a function of average per-clinic admixture proportions. Again, given that you do see population structure, I think it's worthwhile to try to get some handle on how it could be impacting/confounding current conclusions, even though I understand larger datasets would be more ideal to really dig into this.

We look forward to receiving your revised manuscript.

Kind regards,

Indira Govender, MBChB, MMed, FCPHM

Academic Editor

Journal Requirements:

Additional Editor Comments (if provided):

Reviewers' comments:

Reviewer's Responses to Questions

**Comments to the Author**

1. If the authors have adequately addressed your comments raised in a previous round of review and you feel that this manuscript is now acceptable for publication, you may indicate that here to bypass the “Comments to the Author” section, enter your conflict of interest statement in the “Confidential to Editor” section, and submit your "Accept" recommendation.

Reviewer #1: All comments have been addressed

Reviewer #2: (No Response)

2. Does this manuscript meet PLOS Global Public Health’s publication criteria? Is the manuscript technically sound, and do the data support the conclusions? The manuscript must describe methodologically and ethically rigorous research with conclusions that are appropriately drawn based on the data presented.

Reviewer #1: Yes

Reviewer #2: Partly

3. Has the statistical analysis been performed appropriately and rigorously?

Reviewer #1: Yes

Reviewer #2: Yes

4. Have the authors made all data underlying the findings in their manuscript fully available (please refer to the Data Availability Statement at the start of the manuscript PDF file)?

Reviewer #1: No

Reviewer #2: No

5. Is the manuscript presented in an intelligible fashion and written in standard English?

Reviewer #1: Yes

Reviewer #2: Yes

6. Review Comments to the Author

Reviewer #1: (No Response)

Reviewer #2: The authors have made significant attempts at address the concerns raised around the first submission, and I appreciate that a number of aspects of the paper are now clearer. I also appreciate that they have removed the HIV-positive individuals from the controls (to be consistent with the definition of cases). Unfortunately, though, there are still a number of significant problems.

Firstly, the authors insist on describing their controls as population-based controls, which is simply not true. The controls were recruited from health facilities and were “TB suspects”, which presumably means they had symptoms suggestive of TB. This strongly suggests the controls are NOT representative of the general population. Failure to acknowledge this issue leads to problems in the interpretation of the results. As I commented in the original submission, older people are more likely to attend clinics than younger people, and thus one cannot assume the age distribution of the controls to be the same as the age distribution of the general adult population. This makes it dangerous to infer a survivor bias based on the difference in age distribution between cases and controls. The paragraph in lines 91-102 is still confusing and unnecessary.

Secondly, the authors still are not providing a clear motivation for why the genetic data belong in this study. The data don’t say anything about TB risk, so don’t they belong in a separate paper?

Thirdly, the authors have only partially addressed my comment about the unexpected finding that TB risk is lowest in people who are born in towns but migrated to rural areas. It’s unsatisfactory that the authors rely on hypotheses of clinic staff that are neither logical nor supported by evidence. If they really wanted to document the perspectives of clinic staff (as they say in their response letter), I would have expected a related research objective with an appropriate qualitative research methodology.

Fourthly, the authors have removed from the main text the first statistical model (the “Common risk factor model”, i.e. the model without interactions). It’s not clear why this was done – I find it quite difficult to interpret the results of the interaction models without being able to compare against the results of the simpler model without interactions.

Lastly, there is still a lot of explanation and interpretation in the Results section that should rather have gone in the Methods section or others sections. For example, there is much description of the statistical models and the hypotheses that are being tested, whic

---

## [Editor Report · Decision Letter 2]

13 Jun 2024

Strong Effect of Demographic Changes on Tuberculosis­­ Susceptibility in South Africa

PGPH-D-23-02135R2

Dear Mr Oshiomah Oyageshio,

We thank you for your considered responses to all queries raised by the reviewers and are pleased to inform you that your manuscript 'Strong Effect of Demographic Changes on Tuberculosis­­ Susceptibility in South Africa' has been provisionally accepted for publication in PLOS Global Public Health.

Best regards,

Indira Govender

Academic Editor
